# Development of Force Sensor System Based on Tri-Axial Fiber Bragg Grating with Flexure Structure

**DOI:** 10.3390/s22010016

**Published:** 2021-12-21

**Authors:** Dongjoo Shin, Hyeong-U Kim, Atul Kulkarni, Young-Hak Kim, Taesung Kim

**Affiliations:** 1School of Mechanical Engineering, Sungkyunkwan University, Suwon 16419, Korea; skizp1224@skku.edu; 2Department of Plasma Engineering, Korea Institute of Machinery & Materials (KIMM), Daejeon 34103, Korea; guddn418@kimm.re.kr; 3Symbiosis Centre for Nanoscience and Nanotechnology, Symbiosis International (Deemed University), Pune 412115, India; atul.kulkarni@scnn.edu.in; 4Department of Internal Medicine, Asan Medical Center, University of Ulsan College of Medicine, Seoul 05505, Korea; mdyhkim@amc.seoul.kr; 5SKKU Advanced Institute of Nanotechnology (SAINT), Sungkyunkwan University, Suwon 16419, Korea

**Keywords:** fiber Bragg grating (FBG), force sensor system, ANSYS, LabVIEW, wavelength

## Abstract

Fiber Bragg grating (FBG) sensors have an advantage over optical sensors in that they are lightweight, easy to terminate, and have a high flexibility and a low cost. Additionally, FBG is highly sensitive to strain and temperature, which is why it has been used in FBG force sensor systems for cardiac catheterization. When manually inserting the catheter, the physician should sense the force at the catheter tip under the limitation of power (<0.5 N). The FBG force sensor can be optimal for a catheter as it can be small, low-cost, easy to manufacture, free of electromagnetic interference, and is materially biocompatible with humans. In this study, FBG fibers mounted on two different flexure structures were designed and simulated using ANSYS simulation software to verify their sensitivity and durability for use in a catheter tip. The selected flexure was combined with three FBGs and an interrogator to obtain the wavelength signals. To obtain a calibration curve, the FBG sensor obtained data on the change in wavelength with force at a high resolution of 0.01 N within the 0.1–0.5 N range. The calibration curve was used in the force sensor system by the LabVIEW program to measure the unknown force values in real time.

## 1. Introduction

Fiber-optic sensors have various applications owing to their special properties, such as being lightweight, easy to terminate, having a high flexibility, and having a low cost. In the last 20 years, various fiber-optic force sensors have been developed. Bending-type optical fiber sensors can be used to measure various physical quantities, such as strain, voltage, pressure, and temperature [1]. One of the fiber-optic sensors is the Fiber Bragg grating (FBG), which is temperature- and strain-sensitive. It is widely used because of its inherent wavelength response and compatibility with multiple uses.

Based on FBG’s advantages, we applied it in cardiac catheterization—a minimally invasive procedure for cardiac ablation of the heart [2]. Up to now, when performing a cardiac catheterization, physicians attempted to reach a particular location in the heart through blood vessels to examine or treat heart issues [3]. Using this method, as the catheter moves within the patient’s vasculature, the tip and sides of the catheter apply varying forces to the vessel wall [4]. When manually inserting the catheter, the physician should sense the force at the catheter tip under the limitation of power. Precise measurement of contact force is required for the safety of the patient because excessive contact force can cause cardiac perforation or tamponade. Several studies have suggested that ranges between 0.2 and 0.3 N are the best conditions to ensure safety; over 0.4 N, several problems, such as perforation and popping, can occur.

This minimally invasive surgical procedure with a catheter offers many benefits, including reduced trauma to tissue, reduced recovery time, and decreased risk of infection [3]. Despite these advantages, many challenges must be overcome in the development of a miniaturized force sensor for use in catheters [5]. The catheter must be small, low-cost, easy to manufacture, and free of interference to integrate the sensor. In addition, such sensors must be made of biocompatible materials and tolerate sterilization procedures that are essential for use within the human body. In particular, the miniaturization requirements of such sensors are challenging. Several previous studies have shown considerable progress in the miniaturization of sensors for catheter application [6,7], but several such sensors are too costly for large-scale production and thus commercialization [8,9,10,11,12]. The most important concern for a force sensor is the miniaturization process. To integrate a sensor with a catheter, its dimensions are constrained by the catheter diameter (typically less than 2.5 mm) [13,14]. Overall, the FBG can be used as a force sensor on the tip of the catheter to prevent these considerations [15,16,17].

The FBG force sensor can monitor the applied force, and solid sensors or electrical sensors are commonly used in catheter applications [18,19]. Moreover, unlike conventional electric force sensors, they do not cause electromagnetic interference (EMI) with other electronic devices in the operating room [20]. Furthermore, the flexibility of the FBG is optimal because catheters are often adjusted to different locations to fit into the blood vessel [21]. In this study, we used an FBG force sensor with an interrogator and program as a single system for use in catheter operations. In addition, we needed to consider only strain parameters for accuracy; therefore, it was necessary to exclude temperature for cardiac contact force measurement [22].

In this study, we fabricated a tri-axial FBG force sensor at the tip of the catheter to develop a single system, and the relationship between the wavelength and force was obtained as a calibration curve. The calibration curve calculated by the LabVIEW program was used to obtain an unknown force value as the output. Performance was determined by designing and simulating flexures to achieve the desired sensitivity for the target in the 0–0.5 N range. After setting up the FBG force sensor system, we evaluated the force sensing on pig skin in real time to simulate the catheter ablation procedure.

## 2. Materials and Methods

### 2.1. Design of Flexure Sturcture for the Fabrication of the FBG Sensor

Flexure structures convert a force applied to a force sensor along a specific direction into a displacement or strain that can be measured by a transducer. The mechanical properties, size, and shape of the flexure structure determine the sensitivity, accuracy, and directional response of the force sensor [23]. The stiffness of the flexure structures, and consequently the amount of deflection, is determined by the component dimensions and material properties. In Figure 1a, two types of flexure structures (helical and perforated) were designed for force sensing. The stainless steel (SS) flexures have an outer diameter of 2 mm, an inner diameter of 1.6 mm, and a height of 4 mm.

The FBG-based force sensor was developed with a flexure tip and fiber-optics, as shown in Figure 1b. We designed the FBG (FIBERPRO, Inc., Daejeon, Korea) with a wavelength of 1550 nm, a fiber diameter of 0.125 mm, a fiber length of 1.5 m, and polyimide recoating. Three FBGs were attached along the flexure with 120° spacing using resin (NOA68; NORLAND PRODUCTS Inc., Jamesburg, NJ, USA) to achieve force measurements [24,25]. The resin was used to attach the FBG to the lumen and SS tube. In addition, the SS tube was attached with resin to the lumen at the tip. The FBG—a crossed-wire structure with a hollow cavity—was attached to the flexure and lumen. The lumen has a larger radius than the SS tube, which can protect the FBG inside [26]. Additionally, cost being an important factor, the total fabrication of the FBG sensor was less than $200.

### 2.2. FBG Sensor Setup

The FBG force sensor was evaluated using a force measurement setup, as shown in Figure 1c. The FBG sensor was mounted on a linear motion stage (UTS 50CC, NEWPORT, Irvine, CA, USA) to control the vertical direction using a motion controller (ESP301, NEWPORT, Irvine, CA, USA) with LabVIEW software. This allowed a control of the the precise position and various angles of the FBG sensor tip, which is pressed onto the load cell (MR04-025, MARK-10, Copiague, NY, USA) to apply an accurate force in multiple directions. The load cell was connected to a computer to acquire data through a force gauge (M7i, MARK-10, USA). The tri-axial FBG was connected to a three-channel interrogator (IFIS110; FIBERPRO, Inc., Daejeon, Korea) to read the Bragg wavelengths. This interrogator has a program for temperature offset, so the effect of temperature can be neglected because wavelength and strain are the only main parameters for the wavelength of the FBG3. 

## 3. Results

### 3.1. Principle of the FBG for Sensor

The FBG-based sensor normally monitors the wavelength shift from changes in the Bragg signal, which is affected by strain and temperature. The Bragg wavelength, or resonance condition of a grating, is given by Equation (1):λ_B_ = 2 *n*Λ(1)
where *n* is the effective index of the core and Λ is the grating pitch [27]. The narrowband spectral component of the Bragg wavelength is reflected by the grating when a spectral broadband light source is injected into an optical fiber. This spectral component is missing from the transmitted light. The bandwidth of the reflected signal depends on several parameters, especially the grating length, but most sensor applications typically range from 0.05 to 0.3 nm. Perturbations of the grating in the device result in a Bragg wavelength shift, which can be detected in the reflected or transmitted spectrum. The strain response is due to the physical compression, shortening, or elongation of the sensor (and corresponding fractional change in grating pitch) and the changes in the fiber index due to photoelastic effects.

### 3.2. Flexure Structure Stress Distribution and Sensitivity

We used ANSYS software to analyze and evaluate the behavior of the designed flexure structures and determine the amount of deformation under various axial and lateral loads. The designed flexure structure must withstand a force of 0.5 N, which is the endpoint of perforation of the blood vessel—a dangerous accident that may occur during an operation. Therefore, the maximum measurable forces for the axial and lateral loads in ANSYS were 0.5 N. Figure 2a,b shows the conditions of lateral forces of helical and perforated pattern structures with a force direction [28,29]. Similarly, Figure 2c,d shows the axial forces for the helical and perforated pattern structures, respectively. The analyzed results of ANSYS show that the stress of the helical structure is higher than its perforated structure after loading 0.5 N in the force direction. After loading, the maximum stress for the helical structure is 468.30 MPa and 206.46 MPa for lateral and axial force, respectively. In contrast, the case of a perforated structure shows 38.77 MPa and 4.89 MPa for lateral and axial force. When a force of 0.5 N was loaded in two directions for each pattern structure, the helical structure showed more stress in the two directions than the perforated structure; thus, the helical structure has higher sensitivity than the perforated structure. We then tested both structures at three different angles (i.e., 0°, 45°, and 90°), as shown in Figure 2e. The sensitivity of the two structures was calculated by dividing the stress (MPa) by the force (N) for the three angles (0°,45°, and 90°) with a repeatability test (*n* = 3). The sensitivity values of the helical structure were 870, 903, and 351 for the 0°, 45°, and 90° conditions, respectively. In contrast, the sensitivity values of the perforated structure were lower than their helical structures of 530, 693, and 19, respectively. Therefore, comparing the sensitivity reveals 160%, 130%, and 1840% differences between the helical and perforated structures for the three conditions. The 90° condition causes greater bending and deformation of the tube compared to the 0° condition. Based on the ANSYS analysis, the helical structure is selected for its higher sensitivity than that of a perforated structure.

### 3.3. Integration of Force Sensor System with the Calibration Curve

Based on the calibration curve between the normalized wavelength and force, with LabVIEW coding the program, the FBG sensor system is shown in Figure 3a. The system was integrated with a touch screen, motherboard (CPU), the LabVIEW program, and an optical sensing interrogator to which three FBG sensors were connected. Initially, three channels of the FBG sensor read the wavelength every 0.01 N (0.1–0.5 N) (Figure 3b–d); the normalized wavelength output is represented in Equation (2). When the motion controller moved every 0.030 mm to the load cell, 0.01 N was applied to the FBG sensor. The three channels of the FBG sensor exhibited a linear relationship with a similar slope. The resolution and sampling rate of the FBG interrogator were 20 pm and 100 Hz, respectively. In addition, the resolution of the load cell was 0.001 N.
Normalized wavelength output = Δλ/λ_max_(2)
where Δλ is the variation in wavelength and λ_max_ is the wavelength of the maximum force (0.5 N). Under axial loading, the FBG force sensor has calibrated wavelength outputs with a linear working range of 0.1–0.5 N at a 0.01 N resolution. There were three normalized wavelength outputs with an error bar to obtain a high-accuracy calibration curve (*n* = 3). The R^2^ values of the calibrated wavelength output are 0.996, 0.993, and 0.989, respectively. The slopes of the calibrated wavelength outputs for the three channels are 0.021, 0.019, and 0.022, respectively. The standard errors of estimation were 0.017, 0.020, and 0.017 (*p* < 0.0001) for the three channels, respectively. Additionally, the hysteresis values were calculated as 2.01%, 2.04%, and 1.89% for channels 1, 2, and 3, respectively. (Table 1) These results show high repeatability and stability as calibration curves. From the calibration curve, the unknown force can be accurately inferred as a change in wavelength for the FBG, and it is programmed in LabVIEW as a system along with other components (see Figure 3a).

### 3.4. Real-time FBG Sensing Performance

After making a force sensor system, we tested it on pig skin in real time. The FBG force sensor was pressed to the skin using precise motion control, the aim being to simulate the catheter ablation procedure in which the physician manually advances and steers the FBG force sensor against the cardiac wall. When the FBG force sensor pressed the pig skin, the wavelength changed depending on the unknown force as the input value in the system, as shown in Figure 4a. Based on the calibration curve, the change in wavelength was immediately estimated to be the force. The outputs of the force from the calibration curve almost matched the outputs using a commercialized force detector with a load cell, as shown in Figure 4b. Furthermore, the average value of the wavelength of the three channels was inputted to the system because the slopes were similar, and the repeatability was high. Therefore, the force and wavelength were monitored simultaneously and showed high sensitivity.

## 4. Conclusions

In summary, we developed an FBG force sensor system for catheter applications to avoid interference with electronic devices in the operating room. Initially, flexure structures were designed in helical and perforated patterns and simulated using ANSYS to compare stress and sensitivity. Moreover, it was determined experimentally that the helical structure was more sensitive than the perforated structure at 0°, 45°, and 90°. Then, we combined a three-channel FBG with a selected helical structure and lumen to use as a force sensor. To obtain a calibration curve, the FBG sensor obtained data on the change in wavelength with force at a resolution of 0.01 N within the 0.1–0.5 N range. After confirming that the change in the Bragg wavelength was similar in the three FBGs, we implemented a high resolution of 0.01 N within the same range three times, the results showing good repeatability and stability. Consequently, we were able to use LabVIEW to program the force sensor system to use a high linear slope as a calibration curve. The force value could then be obtained in real time from the wavelength change of the FBG force sensor. Our FBG force sensor system has the advantages of high repeatability and stability and is expected to be used by physicians in cardiac catheterization.

## Figures and Tables

**Figure 1 sensors-22-00016-f001:**
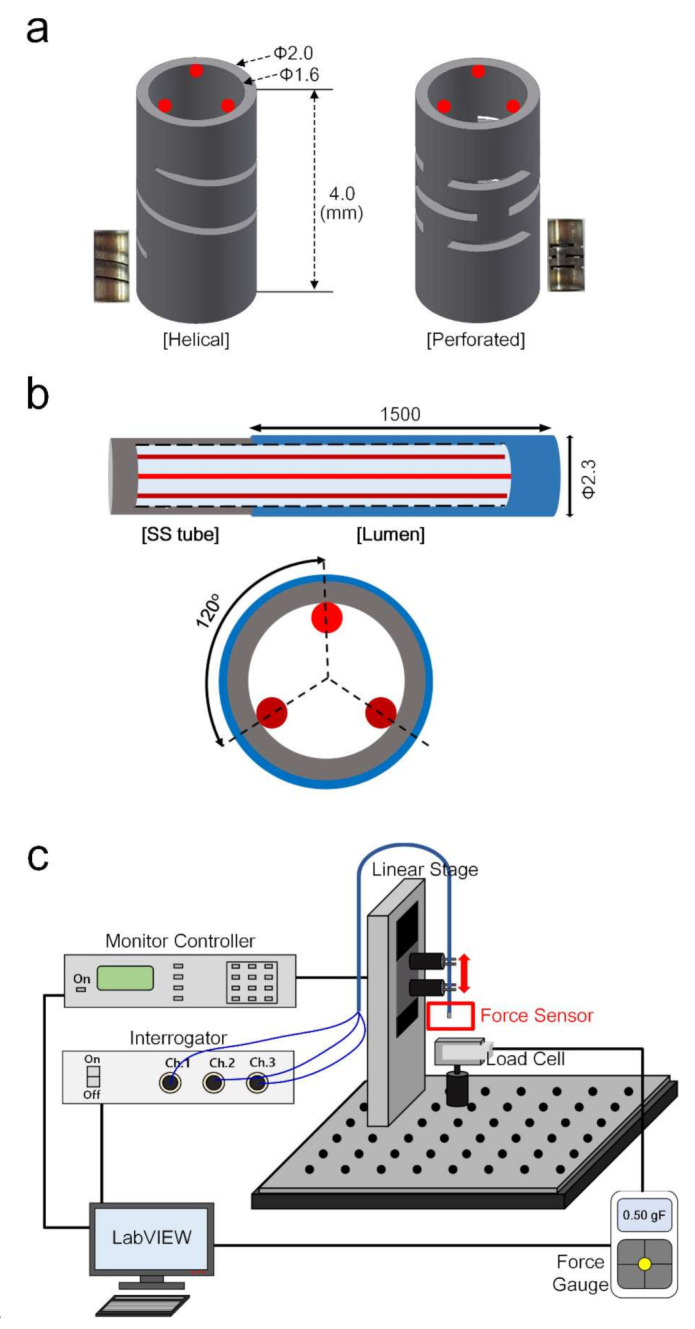
(**a**) Helical and perforated patterned structure for the tip of the force sensor. (**b**) Schematic of a tri-axial FBG force sensor with cross-section (red line is FBG). (**c**) Force measurement setup for the calibration curve of force and wavelength.

**Figure 2 sensors-22-00016-f002:**
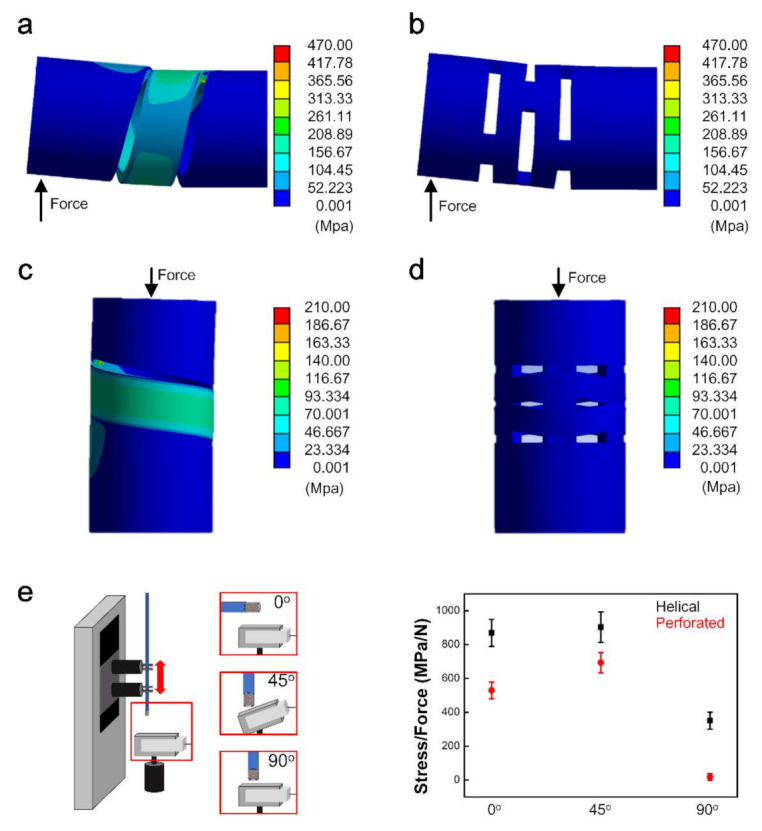
Stress distributions for 0.5 N loading by ANSYS. (**a**) Lateral force analysis for the helical structure and (**b**) perforated pattern structure. (**c**) Axial force analysis for the helical structure; and (**d**) perforated pattern structure. (**e**) Three-degree conditions for force (0, 45, 90°) with the sensitivity of helical and perforated pattern structures (*n* = 3).

**Figure 3 sensors-22-00016-f003:**
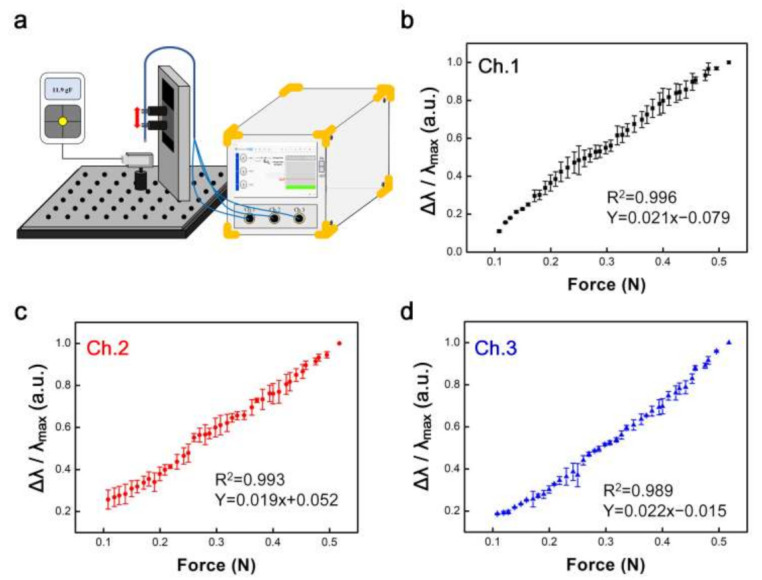
(**a**) Schematic integration of the force sensor system. (**b**–**d**) Calibration curve between normalized wavelength and force at intervals of 0.01 N (*n* = 3).

**Figure 4 sensors-22-00016-f004:**
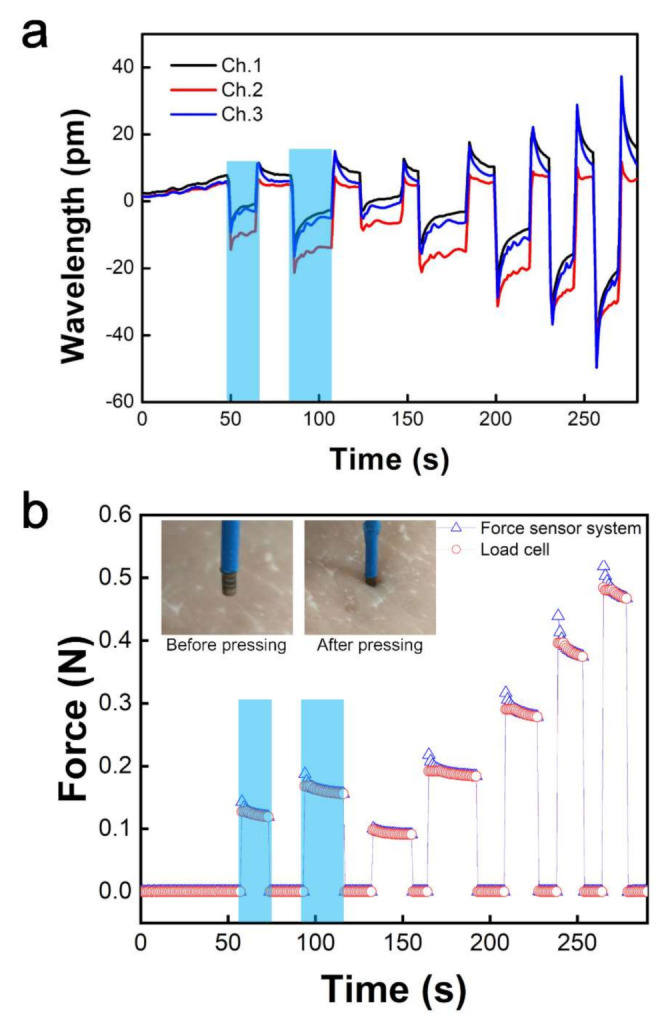
Real-time force sensing on pig skin using the FBG force sensor. (**a**) Wavelength inputted to force sensor system. (**b**) Force graph output from the load cell and the wavelength input value of the force sensor system.

**Table 1 sensors-22-00016-t001:** R square, linearity, standard error, and hysteresis of three FBGs.

	R Square	Linearity	Standard Error	Hysteresis
Ch. 1	0.996	0.021	0.017	2.01%
Ch. 2	0.993	0.019	0.020	2.04%
Ch. 3	0.989	0.022	0.017	1.89%

## Data Availability

The data presented in this study are available on request from the corresponding author.

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
