# Peer review of "Development of Force Sensor System Based on Tri-Axial Fiber Bragg Grating with Flexure Structure"

_sensors, 2021, doi:10.3390/s22010016_

Round 1
Reviewer 1 Report
I have reviewed this manuscript. This paper was published in IEEE conference format with little (or more less no) new contribution. Authors have just added some experiment in their current article that is been submitted for review. At the same time, I found some issues with respect to the study reported in the manuscript and I would suggest authors to supplement their studies with the following:
- [Page1, line24] at a high resolution of 0.01N? The resolution is deduced from the sensitivity matrix obtained in the static test of the designed sensor, and the resolution of the three axes is not indicated.So what does the author mean here by 0.01N?
- [L95] Lumen was not introduced. Different materials have different effects on sensors; what is the bonding mode of FBG? How about FBG sticking to Lumen or stainless steel tube?
- [L107] Please specify the temperature migration procedure.
- [L109] Please provide a detailed experimental entity diagram.
- [L145] Please give a specific experimental diagram, not a simple schematic diagram.
- [L146] Please give the specific applied load. Is the stress obtained in Fig. e based on finite element simulation or specific experiment?
- [L180] Please give the basic characteristic - repeatability index of the sensor in detail.
- [L187] Before the real-time force sensing experiment of pigskin, the author needs to judge the accuracy of the calibration model of the designed sensor. The author only provides the calibration experiment, which cannot verify the accuracy of the sensor in actual use. The author needs to supplement the dynamic load test to judge the precision of the designed sensor by comparing the force value captured by the commercial sensor and the FBG sensor.
Reviewer 2 Report
See attached file.

Author Response
Reviewer #1 (Remarks to the Author): This manuscript presents a tri-axial FBG-based force sensor for catheters. The overall design of the sensor seems novel and feasible. However, significant improvements are needed to make the work publishable. Detailed comments or questions are listed below:
Comment 1: Many similar FBG sensors for catheters and flexible endoscopic surgical robots have been proposed in the literature. A review of existing studies needs to be included in the introduction. In addition, the unique contribution and novelty of this work need to be highlighted. Some suggested references are given below
Li et al., 2018, Three-Dimensional Catheter Distal Force Sensing for Cardiac Ablation Based on Fiber Bragg Grating
Lai et al., 2021, A Three-axial Force Sensor Based on Fiber Bragg Gratings for Surgical Robots
Response 1: Thank you for your suggestion. We added in revised manuscript as ref.10
(Line. 62) “Overall, the FBG can be used as a force sensor on the tip of the catheter to prevent these considerations [10].”
Reference
[10] T. Li; C. Shi; H. Ren, Three-Dimensional Catheter Distal Force Sensing for Cardiac Ablation Based on Fiber Bragg Grating. IEEE ASME Trans Mechatron. 2018, 23, 2316-2327.
Comment 2: The authors compared the perforated tip structure and helical tip structure. Why these two? There are many feasible flexible tip designs. In addition, the perforated tip structure is obviously much stiffer than the helical design and thus has much lower sensitivity. This design could be excluded out even before doing further ANSYS analysis. In addition, for the tip design, sensitivity shall not be the only consideration; strength is another concern, i.e., the tip may have yield or fatigue failure if its stress is too high (although it may have high sensitivity).
Response 2: Thank you for your comments. As you suggestion, we had many options for design but these two structures are close to our goal for protection of FBG and increase sensitivity with flexibility. We considered when the stress is too high, the structure should be protected to FBG optical fiber up to 0.5 N so we confirmed numerical analysis for deformation with two structures at first. We set the 0.5 N as maximum force because it may cause perforation on heart wall or blood vessel.
Comment 3: Three FBGs were used. Which forces they are measuring? How are they decoupled to measure these forces?
Response 3: All three FBGs measured force values simultaneously and compared average of measured wavelength responses to infer the force values applied to the tip. Since the tip of catheter controlled by manually, the contact point will be tip of catheter with detection of force sensor.
Comment 4: How is the hysteresis of the sensors?
Response 4: Sorry for missing information of hysteresis. We calculated the hysteresis values at 2.01%, 2.04%, and 1.89% for channel 1, 2, and 3, respectively. We modified in revised manuscript.
(Line. 175) “Additionally, the hysteresis values were calculated as 2.01%, 2.04%, and 1.89% for channels 1, 2, and 3, respectively.”
Comment 5: How is the FBG bonded to the tip of the catheter?
Response 5: Three FBG were attached along the flexure structure at 120° intervals using resin. In addition, in order to completely fix the FBG, it was attached to the lumen using a resin. (fig.1 a, b)
(Line. 95) “In addition, the SS tube was attached with resin to the lumen as the tip.”
Fig. 1. (a) Helical and perforated patterned structure for the tip of the force sensor (b) schematic of a tri-axial FBG force sensor with cross-section (red line is FBG) (c) force measurement setup for the calibration curve of force and wavelength.
Comment 6: Could the catheter with this sensor design be sterilized?
Response 6: Yes, we could. The catheter product itself is used in hospital for patient and we used it. Moreover, the stainless-steel tube which uses in our force sensor system does not rust and responds harmlessly to the human body. [R1]
[R1] S. Lim; S. Lee; S. Yi; Y. Son; S. Choi; Y. Kim, The Biological Safety of Stainless Steel Needles Used in Warm-needling, Evid. Based Complementary Altern. Med., 2008, 7, 259-264.
Comment 7: The writing must be significantly improved as there are many grammatic errors. For instance, “Overall, considerations, FBG can be used as a force sensor on tip of the catheter to prevent these considerations. Therefore, we preliminary fabricated and tested the possibility as force sensor.”
Response 7: Sorry for our mistakes. We did English correction again by ‘Editage’ overall the revised manuscript.

Reviewer 3 Report
In this manuscript, the authors reported a force sensor system based on tri-axial fiber Bragg grating with flexure structure. The innovation of the work is incremental, and the simulation and experimental results are preliminary. Furthermore, the author fails to convince the reviewer the necessity to use three FBGs in their design because there is no result about the vector force measurement in this manuscript. Therefore, significant enrichment and modifications are required before this manuscript can be published in Sensors.
- What were the length of the FBGs? What are their locations in the flexure?
- Starting from line 124, could the authors provide equations to explicitly describe the strain response of the FBG.
- In Fig. 2(a)-(d), it is better to adjust the color bar to matches the range of the simulation results. Currently the higher limit of the color bar is too large.
- How did the author measure the sensitivity? Was it based on the calibration curve in section 3.3? The authors should clarify it.
- The results in Fig. 3(b) and (c) seems to be redundant since calibration curves with higher resolution are presented in Fig. 3(d)-(f). Do the authors have specific reasons to include those two curves?
- In Fig. 4, the authors present the measurement results based on the calibration curve. To better demonstrate the sensor’s accuracy, the authors should also provide the measurement results from a commercial force sensor as the comparison.
- In line 93, the authors claim that their sensor is to achieve vector force measurements. However, the authors did not report any simulation or experimental result to demonstrate the capability to measure vector force.
Author Response
Remarks to the Author: In this manuscript, the authors reported a force sensor system based on tri-axial fiber Bragg grating with flexure structure. The innovation of the work is incremental, and the simulation and experimental results are preliminary. Furthermore, the author fails to convince the reviewer the necessity to use three FBGs in their design because there is no result about the vector force measurement in this manuscript. Therefore, significant enrichment and modifications are required before this manuscript can be published in Sensors.
Point 1: What were the length of the FBGs? What are their locations in the flexure?
Response 1: Thank you for your comments. As we mentioned in manuscript, the length of the FBG is 1.5 m and diameter of FBG is 0.125 mm. The FBGs were attached with a tip using resin at 120o spacing to end of flexure. The figure 1b shows the red line and points as FBGs.
Fig. 1. (a) Helical and perforated patterned structure for the tip of the force sensor (b) schematic of a tri-axial FBG force sensor with cross-section (red line is FBG) (c) force measurement setup for the calibration curve of force and wavelength.
Point 2: In Fig. 2(a)-(d), it is better to adjust the color bar to matches the range of the simulation results. Currently the higher limit of the color bar is too large.
Response 2: Thank you for your comments. We modified to match the range of results at each condition in Fig.2.
Fig.2 Stress distributions for 0.5 N loading by ANSYS. (a) lateral force analysis for helical structure and (b) perforated pattern structure, (c) axial force analysis for helical structure and (d) perforated pattern structure; (e) three-degree conditions for force (0, 45, 90°) with the sensitivity of helical and perforated pattern structures. (n=3).
Point 3: The author did not detail how they compensate the temperature influence on the wavelength shift of the fiber Bragg gratings.
Response 3: Our interrogator (IFIS110) can set up a measurement environment that is optimized for strain or temperature with our choice. The detail of mechanism is not explained in manual of our interrogator. However, the change of wavelength was insignificant at 25-40° for respecting to strain. Therefore, the temperature of human body and blood vessel about 37° does not seem to be affected to FBG.
Point 4: How did the author measure the sensitivity? Was it based on the calibration curve in section 3.3? The authors should clarify it.
Response 4: The sensitivity was defined as the value of the change in wavelength at each force divided by the change in wavelength at maximum force (0.5 N). According to various literatures on catheters, the range of 15-40 gF (=0.15-0.4 N) is usually referred to as moderate contact forces. If force is applied more than 0.5 N, it causes perforation in the cardiac wall, so we set 0.5 N as the maximum force value.
Point 5: The results in Fig. 3(b) and (c) seems to be redundant since calibration curves with higher resolution are presented in Fig. 3(d)-(f). Do the authors have specific reasons to include those two curves?
Response 5: We agree your opinions, so we modified Fig. 3 by deleting low resolution graph.
Figure.3 (a) Schematic integration of force sensor system. (b-d) Calibration curve between normalized wavelength and force at intervals of 0.1 N.
Point 6: In Fig. 4, the authors present the measurement results based on the calibration curve. To better demonstrate the sensor’s accuracy, the authors should also provide the measurement results from a commercial force sensor as the comparison.
Response 6: Thank you for your comments. We measured both our sensor and commercial force sensor at the same time with high accuracy so did not attach before manuscript. Herein, we added in commercial force sensor result in fig.4c in revised manuscript.
Figure.4 Real-time force sensing on pig skin using the FBG force sensor. (a) Wavelength input to force sensor system. (b) Force graph output from the wavelength input value of the force sensor system. (c) Force graph output from the loadcell.
(Line. 198) “The outputs of the force from the calibration curve almost matched the outputs using a commercialized force detector with a load cell, as shown in figure 4(c).”
Point 7: In line 93, the authors claim that their sensor is to achieve vector force measurements. However, the authors did not report any simulation or experimental result to demonstrate the capability to measure vector force.
Response 7: Thank you for your comments. As your suggestion, it would be better to achieve vector force measurement, but our system is not achieved yet. [R1] However, during the catheter operation, it was difficult to contact laterally tip of the catheter and the direction of contacted part of wall is only vertical. Therefore, our force sensor designed for targeting vertical direction.
[R1] P. Polygerinos; A. Ataollahi; T. Schaeffter; R. Razavi; L. Seneviratne, MRI-Compatible Intensity-Modulated Force Sensor for Cardiac Catheterization Procedure, IEEE. Trans. Biomed., 2011, 58, 721-726.

Reviewer 4 Report
Dear Authors,
the English writing of this paper is very poor and has to be improved significantly before submission.
In your submission, the description of the medical motivation shows major deficits and a mixture of facts that do not fit together. This has to be changed completely. You state, that you want to make cardiac catheterization safer by adding a force sensor to the catheter. For cardiac ablation, the catheter tip has to get in contact with the inner heart surface and, since the heart is beating, for scientific reasons it might be of interest to know the forces that appear. In reality, the catheter is curved in the vasculature, and forces applied at the handpiece are only transmitted in a very limited way to the catheter tip.
Ablation catheters include cables and electrodes for ablation and a bending mechanism for steering of the tip. Does this influence the fiber bragg gratings?
Forces during catheterization appear not only at the tip but also at every contact zone along the catheter and the vascular wall. These forces might be more relevant.
You state a cost of 200$ for the Fibres how is the cost relation between this and the cost of a standard (certified) ablation catheter compared to the value that is added by using the Force sensing?
The active tip of your system has open slits. How is this usable in a real application inside the blood flow? There is a high risk for clotting.
Can ensure that the force system will survive standard sterilization procedures?
How would the fibers be embedded inside a real catheter system?
Perforation is depending on the force applied over a surface. Heart catheters are big in diameter and have atraumatic tips. For which applications are the forces you mentioned?
The experimental setup and protocol are based on the wrong assumptions of your motivation. Please improve.
The key message of the paper is, that you have simulated and tested two types of shapes for the front end of a force sensing system where one system performs better. You state, that this now could be used in the cardiac intervention. Please be honest with yourself.
Author Response
the English writing of this paper is very poor and has to be improved significantly before submission.
In your submission, the description of the medical motivation shows major deficits and a mixture of facts that do not fit together. This has to be changed completely. You state that you want to make cardiac catheterization safer by adding a force sensor to the catheter. For cardiac ablation, the catheter tip has to get in contact with the inner heart surface and, since the heart is beating, for scientific reasons it might be of interest to know the forces that appear. In reality, the catheter is curved in the vasculature, and forces applied at the handpiece are only transmitted in a very limited way to the catheter tip.
Response: Thank you for mentioned comments. Actually, we did government project (HI14C0517) with department of cardiology, Asan medical center. (Republic of Korea). The force sensor is necessary for catheter in real field because commercial catheter does not have force sensor on the tip. Therefore, the surgeon needs to manually control very carefully avoiding perforation so installing force sensor in catheter has very high needs for surgeon. Our sensor is initial step for this problem and difficult to apply real patient yet. Near future, we hope to achieve more advanced force sensor in catheter.
Point 1: Ablation catheters include cables and electrodes for ablation and a bending mechanism for steering of the tip. Does this influence the fiber bragg gratings?
Response 1: Thank you for comments. As you mentioned ablation catheter has many cables and electrode, so we used only side part of cable. The center part is remained for them and FBG is never influenced to cables or electrode also. Additionally, the tip of FBG has a Bragg grating for bending or sensing, so other parts of the FBG are not affected to Bragg grating.
Point 2: Forces during catheterization appear not only at the tip but also at every contact zone along the catheter and the vascular wall. These forces might be more relevant.
Response 2: Thank you for comments. As mentioned Response1, the tip of FBG has a Bragg grating for bending or sensing, so other parts of the FBG are not affected to Bragg grating. It can be ignored for contacting other parts with vascular wall.
Point 3: You state a cost of 200$ for the Fibres how is the cost relation between this and the cost of a standard (certified) ablation catheter compared to the value that is added by using the Force sensing?
Response 3: Thank you for your valuable comments. We mentioned the cost of 200$ which counts only the fabrication of SS tip and optical fiber for connecting commercial catheter. The price of other sensors used in catheter around 500$ [R1] and it also not account for catheter. Therefore, our force sensor price has an advantage in terms of price.
[R1] Z. Dong, A COST ANALYSIS OF PERIPHERALLY INSERTED CENTRAL CATHETER IN PAEDIATRICS, UNIVERSITY OF TORONTO, central.bac-lac.gc.ca, Canada, 2011.
Point 4: The active tip of your system has open slits. How is this usable in a real application inside the blood flow? There is a high risk for clotting.
Response 4: Thank you for your comments. When we designed the force sensor in ablation catheter, the center parts should be remained space for many electrodes, water pipe and cables. After development of force sensor with tip, the catheter will be assembled other parts in hospital for ablation catheter so our system will not have open slits. Furthermore, it is not easy to clot while blood is flowing inside blood vessel, and it doesn’t clot during catheterization. Besides, the optical fiber can measure in wetting condition and the stainless steel which is often used for making medical tools also doesn’t rust during catheterization. [R2]
[R2] N. Solomon; I. Solomon, Effect of deformation-induced phase transformation on AISI 316 stainless steel corrosion resistance, Eng. Fail. Anal., 2017, 79, 865-875.
Point 5: Can ensure that the force system will survive standard sterilization procedures?
Response 5: Yes, we can. The catheter product itself is used in hospital for patient and we used it. Moreover, the stainless-steel tube which uses in our force sensor system does not rust and responds harmlessly to the human body.
[R3] S. Lim; S. Lee; S. Yi; Y. Son; S. Choi; Y. Kim, The Biological Safety of Stainless Steel Needles Used in Warm-needling, Evid. Based Complementary Altern. Med., 2008, 7, 259-264.
Point 6: How would the fibers be embedded inside a real catheter system?
Response 6: The diameter of lumen is 2.3 mm, which is about 20 times larger than the FBGs we used (optical fiber 0.125 mm), and other materials do not interfere with the embedding of the optical fiber. This SS structure is initially designed the space of other parts (cables, electrodes, water pipe) with surgeon in Asan medical center.
Point 7: Perforation is depending on the force applied over a surface. Heart catheters are big in diameter and have atraumatic tips. For which applications are the forces you mentioned?
Response 7: Thank you for your comments. When we designed the force sensor of catheter, we applied to commercial products of catheter, so size and diameter are maintained. We only replaced the tip for installing force sensor with FBG. The force sensor will be used to measure how much force is applied to the cardiac wall when the catheter tip is applied to surface. Based on this measured force, it helps the surgeon adjust avoid perforation during catheterization.
Point 8: The experimental setup and protocol are based on the wrong assumptions of your motivation. Please improve.
Response 8: Thank you for comments. We assumed that the change of wavelength is correlated with the force using a load cell, and the force value is inferred from the wavelength change into a calibration curve without knowing the force. Therefore, we can get force (Fig.4b) by change of wavelength (Fig.4a) with calibration curve and it is confirmed again with load cell in Fig.4c.
Figure.4 Real-time force sensing on pig skin using the FBG force sensor. (a) Wavelength input to force sensor system. (b) Force graph output from the wavelength input value of the force sensor system. (c) Force graph output from the loadcell.
(Line. 198) “The outputs of the force from the calibration curve almost matched the outputs using a commercialized force detector with a load cell, as shown in figure 4(c).”
Point 9: The key message of the paper is, that you have simulated and tested two types of shapes for the front end of a force sensing system where one system performs better. You state that this now could be used in the cardiac intervention. Please be honest with yourself.
Response 9: Thank you for your opinion. This works is started from government project with department of cardiology, Asan medical center. (HI14C0517) Therefore, we discussed to needs of surgeon and how to apply force sensor in ablation catheter for last 5 years. We designed to attach FBG side part because the center part is need to put other electrodes and cables as needs. This force sensor is initial setup for developing force sensor catheter in Korea so system is not mature yet but will be applied to patient.

Round 2
Reviewer 1 Report
Remarks to the Author: In this manuscript, the authors reported a FBG-based catheter force sensor prototype for cardiac intervention. However, the design, simulation, and current experimental results requires several modifications, which has been suggested below before I would be able to make a recommendation.
Comments:
- Within the Abstract section, the authors mentioned, “the physician should sense the force at the catheter tip under the limitation of power (> 0.5 N)”, what does the authors meant by this statement. The allowable tool-vessel contact force during PCI is 0.5 N while tool-tissue contact force is 0.2 - 0.4 N. In none of these cases does it exceed 0.5 N. In addition, what does the limitation of power mean; the grammatical usage should be rechecked?
- Similarly, in the Abstract section, the authors made a claim that their FBG force sensor is low-cost. However, it is a general knowledge that the FBG technology is an expensive one, while the cost of fabricating your flexure might be low; the cost of purchasing the load cell and the FBG interrogator is high. Hence, the authors should remove such claim from their manuscript.
- “The selected flexure was combined with three channels of the FBG and interrogator” within the Abstract. There is nothing like three channels of the FBG, instead should be “….. with three FBGs and an interrogator”.
- Second paragraph under the Introduction section, “Based on FBG’s advantages, we applied it to a catheter, which is a minimally invasive procedure for …. (PCI) of the heart”. This statement is erroneous; the catheter is only a tool used during minimally invasive procedure and not as stated in the manuscript. As such, the statement should be reworded appropriately. In addition, the authors made reference to PCI but later made reference to cardiac ablation in the manuscript, thus the authors should use words incorporating either PCI and Cardiac Ablation or stick to cardiac ablation within the Introduction section.
- The authors have not carried out an extensive review of existing literature on FBG-based sensors for cardiac intervention as well as comparing the FBG sensors with other approaches such as electrical-based and image-based. Hence, the start-of-the-art within the manuscript is weak. The following existing works are strongly suggested to be discussed within the Introduction section. In Line 57, third paragraph under the Introduction section, the authors mentioned, “several previous studies have shown considerable progress in the miniaturization of sensors for catheter application”. Like mentioned above this should be properly discussed highlighting the strengths and weaknesses with the listed references a good guide. Finally, a clear contribution of this study should be added.
- Shin, T. Kim, and H. U. Kim, ‘Development of Tri-axial Fiber Bragg Grating Force Sensor in Catheter Application’, MeMeA 2018 - 2018 IEEE Int. Symp. Med. Meas. Appl. Proc., pp. 1–5, 2018.
- Gao, Y. Zhou, L. Cao, Z. Wang, and H. Liu, ‘Fiber Bragg Grating-Based Triaxial Force Sensor with Parallel Flexure Hinges’, IEEE Trans. Ind. Electron., vol. 65, no. 10, pp. 8215–8223, 2018.
- Taghipour, A. N. Cheema, X. Gu, and F. Janabi-Sharifi, ‘Temperature Independent Triaxial Force and Torque Sensor for Minimally Invasive Interventions’, IEEE/ASME Trans. Mechatronics, vol. 25, no. 1, pp. 449–459, Feb. 2020.
- O. Akinyemi et al., "Development of a Millinewton FBG-Based Distal Force Sensor for Intravascular Interventions," 2020 16th International Conference on Control, Automation, Robotics and Vision (ICARCV), Shenzhen, China, 2020. T. O. Akinyemi et al., "Fiber Bragg Grating-Based Force Sensing in Robot-Assisted Cardiac Interventions: A Review," IEEE Sensors Journal, 21(9), pp. 10317-31, 2021.
- J. Payne, H. Rafii-Tari, and G. Z. Yang, ‘A force feedback system for endovascular catheterisation’, IEEE Int. Conf. Intell. Robot. Syst., pp. 1298–1304, 2012, DOI: 10.1109/IROS.2012.6386149.
- J. Pandya, J. Sheng, and J. P. Desai, ‘Towards a tri-axial flexible force sensor for catheter contact force measurement’, Proc. IEEE Sensors, no. Mask 1, pp. 1–3, 2017, DOI: 10.1109/ICSENS.2016.7808954.
- Noh et al., “Image-based optical miniaturized three-axis force sensor for cardiac catheterization,” IEEE Sensors J., vol. 16, no. 22, pp. 7924–932, Nov. 2016.
- Atieh, R. Ahmadi, M. Kalantari, J. Dargahi, and M. Packirisamy, ‘A piezoresistive based tactile sensor for use in minimally invasive surgery’, 2011 IEEE 37th Annu. Northeast Bioeng. Conf. NEBEC 2011, pp. 3–4, 2011, DOI:10.1109/NEBC.2011.5778607.
- Su et al., ‘Fiber-Optic Force Sensors for MRI-Guided Interventions and Rehabilitation: A Review’, IEEE Sens. J., vol. 17, no. 7, pp. 1952–1963, Apr. 2017, DOI:10.1109/JSEN.2017.2654489.
- Under section 2.1, Line 88, the author mentioned “ two types of flexure structures were designed to apply force sensors” I guess the authors meant force sensing instead.
- Did the authors considered the alternative use of 3D-printed materials (which is a better biocompatible material) during their flexure design
- The authors mentioned that the FBG was manufactured in “FIBERPRO” what does this mean. Is this the name of the company where the optical fiber with FBG was bought? I am not sure if this information is essential within the manuscript.
- The resolution and sampling rate of the FBG interrogator and load cell should be mentioned in the manuscript.
- The claim about temperature offset by the interrogator is debatable, such claim should be backed up with a reference or verifiable proof
- I would like to know how convenient it was for the authors to deal with the miniature dimension of the sensor, did the authors considered increasing the length of the sensor to about 10-15mm?
- Although the allowable tool-vessel contact force is within 0.2-0.4 N, the authors should show that their sensor design can accommodate a larger force values closer to 1.0 N. Currently, are the authors suggesting that the sensor could break if a force beyond 0.5 N is applied. For factor of safety consideration, the numerical simulation of the sensor in ANSYS should be reported for force values between 0.7 – 1.0 N. In addition, the author should include a table, outlining parameters (like Young modulus, Poisson ratio, and Density) of the materials used for the FEM-Based simulation. Furthermore, the authors did not perform a simulation of the assembled sensor component comprising of the flexure, adhesive, and fiber assembly, this simulation result should be added to the manuscript.
- The authors reported about the stress distribution of the flexure as simulated in ANSYS, the deformation and normal elastic or equivalent strain values of the sensor may also be presented within the manuscript.
- Did the authors considered using four fibers and hence been able to compute the two-dimensional force measurement of the sensor using two fibers per axis e.g 1-2 ( X-axis) and 3-4 (Y-axis), this would mean the fibers placed at 90 degrees to each other, and could facilitate temperature effect decoupling instead of the claim made in the paper.
- In section 3.3, equation (2) was mentioned instead of (1), also Figure 3(c) was mentioned in Line 171 but the figure is missing, the current Fig. 3c is the one mentioned in Line 182
- The authors are encouraged to include a table highlighting the sensor dimension, sensitivity, linearity errors and R square values.
- From Figure (3), the average sensitivity of the sensor is 50pm/N, this is moderately low, do the authors think that the arrangement of the fibers at 120 degrees from each other contributes to this effect?
- Figure 4b and c should be combined into one figure such that its easier to see the measurement and the deviations in the readings of the two sensors
- Statement about vector force measurement should be removed from the manuscript as the authors did not sufficiently justify that in the current version of the manuscript.
- Did the authors considered other flexure designs apart from the hollow shape utilized in this manuscript; the following articles might be helpful as well as some mentioned earlier above.
- A. Turkkan, V. K. Venkiteswaran, and H. Su, ‘Rapid conceptual design and analysis of spatial flexure mechanisms’, Mech. Mach. Theory, vol. 121, pp. 650–668, 2018, doi: 10.1016/j.mechmachtheory.2017.11.025.
Li, L., Zhang, D., Guo, S., & Qu, H. A generic compliance modeling method for two-axis elliptical-arc-filleted flexure hinges. Sensors (Switzerland), 17(9), 1–20, 2017, doi.org/10.3390/s17092154
- Shi, T. Li, and H. Ren, ‘A Millinewton Resolution Fiber Bragg Grating-Based Catheter Two-Dimensional Distal Force Sensor for Cardiac Catheterization’, IEEE Sens. J., vol. 18, no. 4, pp. 1539–1546, 2018, DOI: 10.1109/JSEN.2017.2779153.
Author Response
Response to Reviewer 1 Comments
Remarks to the Author: In this manuscript, the authors reported a FBG-based catheter force sensor prototype for cardiac intervention. However, the design, simulation, and current experimental results requires several modifications, which has been suggested below before I would be able to make a recommendation.
Point 1: Within the Abstract section, the authors mentioned, “the physician should sense the force at the catheter tip under the limitation of power (> 0.5 N)”, what does the authors meant by this statement. The allowable tool-vessel contact force during PCI is 0.5 N while tool-tissue contact force is 0.2 - 0.4 N. In none of these cases does it exceed 0.5 N. In addition, what does the limitation of power mean; the grammatical usage should be rechecked?
Response 1: Thank you for your comments. Sorry for the wrong expression so we modified the sentence in abstract. The sentence means that the limitation of power should be lower than 0.5 N and the case of allowable tool-vessel contact force during PCI is 0.5 N while tool-tissue contact force is 0.2 - 0.4 N. Therefore, we modified in revised manuscript.
[Line 20] “When manually inserting the catheter, the physician should sense the force at the catheter tip under the limitation of power (< 0.5 N)”
Point 2: Similarly, in the Abstract section, the authors made a claim that their FBG force sensor is low-cost. However, it is a general knowledge that the FBG technology is an expensive one, while the cost of fabricating your flexure might be low; the cost of purchasing the load cell and the FBG interrogator is high. Hence, the authors should remove such claim from their manuscript.
Response 2: Thank you for your comments. As you know, load cell and FBG interrogator are expensive, but we mentioned FBG force sensor only. Since the load cell is initially needed for getting calibration curve, the FBG force sensor doesn’t need it for operation. As shown in Figure R1, the FBG force sensor is combined with catheter, FBG and stainless steel (SS) structure. The detail of cost is only to the cost of adding a force sensor to the catheter, not the total cost required for the catheter procedure. The figure R1(a) shows real catheter for using operation in the hospital and its cost is expensive. However, we applied to modify the tip parts with SS structure and FBG for force sensor. We mentioned it for cost $200 in manuscript.
Figure R1. (a) The commercial catheter with a length of 1.5 m, (b) the full length of 1.5 m fiber with FBG sensing part, (c) the FBG sensing part.
Point 3: “The selected flexure was combined with three channels of the FBG and interrogator” within the Abstract. There is nothing like three channels of the FBG, instead should be “….. with three FBGs and an interrogator”.
Response 3: Thank you for your comments. We modified the sentence in revised manuscript.
[line 24] “The selected flexure was combined with three FBGs and interrogator to obtain the wavelength signals.”
Point 4: Second paragraph under the Introduction section, “Based on FBG’s advantages, we applied it to a catheter, which is a minimally invasive procedure for …. (PCI) of the heart”. This statement is erroneous; the catheter is only a tool used during minimally invasive procedure and not as stated in the manuscript. As such, the statement should be reworded appropriately. In addition, the authors made reference to PCI but later made reference to cardiac ablation in the manuscript, thus the authors should use words incorporating either PCI and Cardiac Ablation or stick to cardiac ablation within the Introduction section.
Response 4: Thank you for your comments. We agree to your comments so remove PCI and only mentioned cardiac ablation in revised manuscript.
[line 40] “Based on FBG’s advantages, we applied it to a catheter, which is a manually invasive procedure for cardiac ablation of the heart [2].”
Point 5: The authors have not carried out an extensive review of existing literature on FBG-based sensors for cardiac intervention as well as comparing the FBG sensors with other approaches such as electrical-based and image-based. Hence, the start-of-the-art within the manuscript is weak. The following existing works are strongly suggested to be discussed within the Introduction section. In Line 57, third paragraph under the Introduction section, the authors mentioned, “several previous studies have shown considerable progress in the miniaturization of sensors for catheter application”. Like mentioned above this should be properly discussed highlighting the strengths and weaknesses with the listed references a good guide. Finally, a clear contribution of this study should be added.
- Shin, T. Kim, and H. U. Kim, ‘Development of Tri-axial Fiber Bragg Grating Force Sensor in Catheter Application’, MeMeA 2018 - 2018 IEEE Int. Symp. Med. Meas. Appl. Proc., pp. 1–5, 2018.
- Gao, Y. Zhou, L. Cao, Z. Wang, and H. Liu, ‘Fiber Bragg Grating-Based Triaxial Force Sensor with Parallel Flexure Hinges’, IEEE Trans. Ind. Electron., vol. 65, no. 10, pp. 8215–8223, 2018.
- Taghipour, A. N. Cheema, X. Gu, and F. Janabi-Sharifi, ‘Temperature Independent Triaxial Force and Torque Sensor for Minimally Invasive Interventions’, IEEE/ASME Trans. Mechatronics, vol. 25, no. 1, pp. 449–459, Feb. 2020.
- Akinyemi et al., "Development of a Millinewton FBG-Based Distal Force Sensor for Intravascular Interventions," 2020 16th International Conference on Control, Automation, Robotics and Vision (ICARCV), Shenzhen, China, 2020.
- O. Akinyemi et al., "Fiber Bragg Grating-Based Force Sensing in Robot-Assisted Cardiac Interventions: A Review," IEEE Sensors Journal, 21(9), pp. 10317-31, 2021.
- Payne, H. Rafii-Tari, and G. Z. Yang, ‘A force feedback system for endovascular catheterisation’, IEEE Int. Conf. Intell. Robot. Syst., pp. 1298–1304, 2012, DOI: 10.1109/IROS.2012.6386149.
- Pandya, J. Sheng, and J. P. Desai, ‘Towards a tri-axial flexible force sensor for catheter contact force measurement’, Proc. IEEE Sensors, no. Mask 1, pp. 1–3, 2017, DOI: 10.1109/ICSENS.2016.7808954.
- Noh et al., “Image-based optical miniaturized three-axis force sensor for cardiac catheterization,” IEEE Sensors J., vol. 16, no. 22, pp. 7924–932, Nov. 2016.
- Atieh, R. Ahmadi, M. Kalantari, J. Dargahi, and M. Packirisamy, ‘A piezoresistive based tactile sensor for use in minimally invasive surgery’, 2011 IEEE 37th Annu. Northeast Bioeng. Conf. NEBEC 2011, pp. 3–4, 2011, DOI:10.1109/NEBC.2011.5778607.
- Su et al., ‘Fiber-Optic Force Sensors for MRI-Guided Interventions and Rehabilitation: A Review’, IEEE Sens. J., vol. 17, no. 7, pp. 1952–1963, Apr. 2017, DOI:10.1109/JSEN.2017.2654489.
Response 5: Thank you for your suggestions, we added the references as below in Introduction section.
[line 57] “Several previous studies have shown considerable progress in the miniaturization of sensors for catheter application [6,7], but several such sensors have become costly for large-scale production, and thus, commercialization [8-12].”
[line 62] “Overall, the FBG can be used as a force sensor on the tip of the catheter to prevent these considerations [15-17].”
[line 64] “The FBG force sensor can monitor the applied force, and solid sensors or electrical sensors are commonly used in catheter applications [18, 19].”
Point 6: Under section 2.1, Line 88, the author mentioned “two types of flexure structures were designed to apply force sensors” I guess the authors meant force sensing instead.
Response 6: Thank you for your suggestion, we modified the sentence in revised manuscript.
[line 87] “In Figure 1(a), two types of flexure structures (helical and perforated) were designed for force sensing.”
Point 7: Did the authors considered the alternative use of 3D-printed materials (which is a better biocompatible material) during their flexure design
Response 7: Thank you for your comments. As you mentioned, 3D-printed materials can be used but it should be metal based 3D-printer. Since we tried to use normal 3D-printer with thermoplastic poly-urethane (TPU), the structure was broken with force. Moreover, the metal 3D printer cannot achieve the precise structure with millimeter (mm) scale. Therefore, we tried to manufacture SS tube for mm scale, and it is cheaper than 3D print method.
Point 8: The authors mentioned that the FBG was manufactured in “FIBERPRO” what does this mean. Is this the name of the company where the optical fiber with FBG was bought? I am not sure if this information is essential within the manuscript.
Response 8: Sorry for our mistake. We need to mention detail of company and product information. FIBERPRO is the company of FBG. As your suggestion, we modified other products also in revise manuscript.
[line 91] “We designed the FBG (FIBERPRO, Inc., South Korea) with a wavelength of 1550 nm, a fiber diameter of 0.125 mm, a fiber length of 1.5 m, and polyimide recoating.”
[line 93] “. Three FBGs were attached along the flexure with 120° spacing using resin (NOA68; NORLAND PRODUCTS Inc., USA) to achieve force measurements [24, 25].”
[line 102] “The FBG sensor was mounted on a linear motion stage (UTS 50CC, NEWPORT, USA) to control the vertical direction using a motion controller (ESP301, NEWPORT, USA) with LabVIEW software.”
[line 104] “It can control the accurate position and various angles of the FBG sensor tip, which is pressed on the load cell (MR04-025, MARK-10, USA) to apply an accurate force in multiple directions.”
[line 106] “The load cell was connected to a computer to acquire data through a force gauge (M7i, MARK-10, USA).”
[line 107] “The tri-axial FBG was connected to a three-channel interrogator (IFIS110; FIBERPRO, Inc., South Korea) to read the Bragg wavelengths.”
Point 9: The resolution and sampling rate of the FBG interrogator and load cell should be mentioned in the manuscript.
Response 9: Thank you for your comments. We added the information of the resolution and sampling rate of FBG interrogator and load cell in revised manuscript.
[line 169] “The resolution and sampling rate of the FBG interrogator were 20 pm and 100 Hz, respectively. In addition, the resolution of load cell was 0.001 N.”
Point 10: The claim about temperature offset by the interrogator is debatable, such claim should be backed up with a reference or verifiable proof.
Response 10: Thank you for your comments. We asked the company for getting any proof of our interrogator. When we measured it, strain and temperature measurement functions can be selected for measurement. If you select strain mode, the result shows only for the effect of strain and the temperature effect are negligible at room temperature (18-25 oC). Although human body temperature (35-38 oC) is slightly higher than room temperature, it is kept constant, so the effect of temperature can be neglected by effect of strain.
Point 11: I would like to know how convenient it was for the authors to deal with the miniature dimension of the sensor, did the authors considered increasing the length of the sensor to about 10-15mm?
Response 11: Thank you for your comments. When we designed the structure, we discussed with surgeon and real product of catheter, so the size and length are fixed. Our force sensor should be fitted inside of catheter and diameter of tube. It was big challenge for us. If the size is bigger than this, it would be easier than now.
Point 12: Although the allowable tool-vessel contact force is within 0.2-0.4 N, the authors should show that their sensor design can accommodate a larger force values closer to 1.0 N. Currently, are the authors suggesting that the sensor could break if a force beyond 0.5 N is applied. For factor of safety consideration, the numerical simulation of the sensor in ANSYS should be reported for force values between 0.7 – 1.0 N. In addition, the author should include a table, outlining parameters (like Young modulus, Poisson ratio, and Density) of the materials used for the FEM-Based simulation. Furthermore, the authors did not perform a simulation of the assembled sensor component comprising of the flexure, adhesive, and fiber assembly, this simulation result should be added to the manuscript.
Response 12: Thank you for your comments. As your suggestions, we simulated stress and deformation distributions for two flexure structures with 0.7 and 1 N axial and lateral loading. The maximum stress and deformation values of the helical were much higher than perforated at all parameters. Moreover, as considering the simulation results, every structure didn’t break with 0.7-1 N conditions. Therefore, we can focus on 0.5 N which is main target force and compared to two structures for sensitivity. Furthermore, we performed the simulation of the helical structure assembled three FBG in figure R4. The maximum stress and deformation for 0.5 N axial loading were 29.365 MPa and 0.2278 mm, respectively. Therefore, the results show much lower stress and deformation than both helical and perforated structure. Although stress and deformation values of helical structure assembled with three FBG decreased, the values were similar to the those of perforated structure, so we used it.
Table R1. The maximum stress and deformation values for 1, 0.7, 0.5 N lateral and axial loading at helical and perforated structures.
|
|
|
1 N |
0.7 N |
0.5 N |
|||
|
|
|
Stress (MPa) |
Deformation (mm) |
Stress (MPa) |
Deformation (mm) |
Stress (MPa) |
Deformation (mm) |
|
Helical |
Lateral |
304.32 |
117.78 |
434.74 |
75.594 |
468.30 |
55.751 |
|
Axial |
200.34 |
32.474 |
286.20 |
22.692 |
206.46 |
15.373 |
|
|
Perforated |
Lateral |
108.56 |
1.5326 |
155.09 |
0.1142 |
38.770 |
0.7255 |
|
Axial |
34.220 |
1.6320 |
48.886 |
1.1424 |
4.8900 |
0.0772 |
|
Figure R2. Stress distributions for 1 N axial loading at (a) helical and (b) perforated structures, and lateral loading at the (c) helical and (d) perforated structures. Deformation distributions for 1 N axial loading at (e) helical and (f) perforated structures, and lateral loading at the (g) helical and (h) perforated structures.
Figure R3. Stress distributions for 0.7 N axial loading at (a) helical and (b) perforated structures, and lateral loading at the (c) helical and (d) perforated structures. Deformation distributions for 1 N axial loading at (e) helical and (f) perforated structures, and lateral loading at the (g) helical and (h) perforated structures.
Figure R4. The (a) stress and (b) deformation distribution for 0.5 N axial loading at the helical structure with three FBGs.
Point 13: The authors reported about the stress distribution of the flexure as simulated in ANSYS, the deformation and normal elastic or equivalent strain values of the sensor may also be presented within the manuscript.
Response 13: Thank you for your comments. We added another force conditions (0.7, 1 N) with both axial and lateral directions. When the conditions of forces are applied to axial direction, the maximum deformation for helical structure and perforated structures were 22.692 mm and 1.6320 mm, respectively (Table R1, Figure R2, R3).
Table R1. The maximum stress and deformation values for 1, 0.7, 0.5 N lateral and axial loading at helical and perforated structures.
|
|
|
1 N |
0.7 N |
0.5 N |
|||
|
|
|
Stress (MPa) |
Deformation (mm) |
Stress (MPa) |
Deformation (mm) |
Stress (MPa) |
Deformation (mm) |
|
Helical |
Lateral |
304.32 |
117.78 |
434.74 |
75.594 |
468.30 |
55.751 |
|
Axial |
200.34 |
32.474 |
286.20 |
22.692 |
206.46 |
15.373 |
|
|
Perforated |
Lateral |
108.56 |
1.5326 |
155.09 |
0.1142 |
38.770 |
0.7255 |
|
Axial |
34.220 |
1.6320 |
48.886 |
1.1424 |
4.8900 |
0.0772 |
|
Figure R2. Stress distributions for 1 N axial loading at (a) helical and (b) perforated structures, and lateral loading at the (c) helical and (d) perforated structures. Deformation distributions for 1 N axial loading at (e) helical and (f) perforated structures, and lateral loading at the (g) helical and (h) perforated structures.
Figure R3. Stress distributions for 0.7 N axial loading at (a) helical and (b) perforated structures, and lateral loading at the (c) helical and (d) perforated structures. Deformation distributions for 1 N axial loading at (e) helical and (f) perforated structures, and lateral loading at the (g) helical and (h) perforated structures.
Point 14: Did the authors considered using four fibers and hence been able to compute the two-dimensional force measurement of the sensor using two fibers per axis e.g 1-2 (X-axis) and 3-4 (Y-axis), this would mean the fibers placed at 90 degrees to each other and could facilitate temperature effect decoupling instead of the claim made in the paper.
Response 14: Thank you for your opinion. That is good idea, but our FBG interrogator can connect only 3 channels, so it is impossible to evaluate force sensor using 4 FBGs in this study.
Point 15: In section 3.3, equation (2) was mentioned instead of (1), also Figure 3(c) was mentioned in Line 171 but the figure is missing, the current Fig. 3c is the one mentioned in Line 182
Response 15: Sorry for our mistake, we modified in the revised manuscript correctly.
[line 165] “Initially, three channels of the FBG sensor read the wavelength every 0.01 N (0.1-0.5 N) in Figure 3(b-d), and it is the normalized wavelength output in equation (2).”
Point 16: The authors are encouraged to include a table highlighting the sensor dimension, sensitivity, linearity errors and R square values.
Response 16: Thank you for your comments. We added the table 1 in revised manuscript.
[line 179] “Additionally, the hysteresis values were calculated as 2.01%, 2.04%, and 1.89% for channels 1, 2, and 3, respectively. (Table 1)”
Table 1. R square, linearity, standard error, and hysteresis of three FBGs
Point 17: From Figure (3), the average sensitivity of the sensor is 50 pm/N, this is moderately low, do the authors think that the arrangement of the fibers at 120 degrees from each other contributes to this effect?
Response 17: Not only the sensitivity of each is low, but also the force sensor system focused on further reducing the error by using the three FBGs’ average values.
Point 18: Figure 4b and c should be combined into one figure such that its easier to see the measurement and the deviations in the readings of the two sensors
Response 18: Thank you for your advice. We combined the figure 4b and 4c for easy to comparison in revised manuscript.
Figure 4. Real-time force sensing on pig skin using the FBG force sensor. (a) Wavelength input to force sensor system. (b) Force graph output from the load cell and the wavelength input value of the force sensor system.
[line 197] “The outputs of the force from the calibration curve almost matched the outputs using a commercialized force detector with a load cell, as shown in Figure 4(b).”
Point 19: Statement about vector force measurement should be removed from the manuscript as the authors did not sufficiently justify that in the current version of the manuscript.
Response 19: Thank you for your advice. We agreed with the reviewer’s comment so deleted it in revised manuscript.
Point 20: Did the authors considered other flexure designs apart from the hollow shape utilized in this manuscript; the following articles might be helpful as well as some mentioned earlier above.
- Turkkan, V. K. Venkiteswaran, and H. Su, ‘Rapid conceptual design and analysis of spatial flexure mechanisms’, Mach. Theory, vol. 121, pp. 650–668, 2018, doi: 10.1016/j.mechmachtheory.2017.11.025.
- Li, L., Zhang, D., Guo, S., & Qu, H. A generic compliance modeling method for two-axis elliptical-arc-filleted flexure hinges. Sensors (Switzerland), 17(9), 1–20, 2017, doi.org/10.3390/s17092154
- Shi, T. Li, and H. Ren, ‘A Millinewton Resolution Fiber Bragg Grating-Based Catheter Two-Dimensional Distal Force Sensor for Cardiac Catheterization’, IEEE Sens. J., vol. 18, no. 4, pp. 1539–1546, 2018, DOI: 10.1109/JSEN.2017.2779153.
Response 20: Thank you for your advice. We added references in revised manuscript.
[line 83] “The mechanical properties, size, and shape of the flexure structure determine the sensitivity, accuracy, and directional response of the force sensor [23].”
[line 135] “Figures 2(a) and 2(b) show the conditions of lateral forces of helical and perforated pattern structures with a force direction [28, 29].”

Reviewer 3 Report
Thanks for the author’s response. However, the manuscript still has flaws and questions.
- In response 1, the authors indicated that the FBG is 1.5 m. Usually a FBG is within few centimeters long, and why do the authors require such a long FBG? Furthermore, based on the simulation results in Fig. 2, the stress is concentrated in a few millimeters section which is much shorter than the length of the FBG. In this case, one should expect distortion of the FBG reflection spectrum due to intra-FBG inhomogeneity. Did the author observe any effects on the measurement results?
- The results in the new Fig.2(a-d) are very different from the previous one under the same simulation setting (0.5 N). For example, in the new Fig. 2a, the maximum stress is around 300 MPa but in the previous version the maximum stress was around 20000 MPa. The author should double check the results in Fig. 2 and make sure they are correct.
- They authors did not answer why they need to use three FBGs since the sensor is targeting vertical direction.
Author Response
Response to Reviewer 3 Comments
Remarks to the Author: Thanks for the author’s response. However, the manuscript still has flaws and questions.
Point 1: In response 1, the authors indicated that the FBG is 1.5 m. Usually, a FBG is within a few centimeters long, and why do the authors require such a long FBG? Furthermore, based on the simulation results in Fig. 2, the stress is concentrated in a few millimeters section which is much shorter than the length of the FBG. In this case, one should expect distortion of the FBG reflection spectrum due to intra-FBG inhomogeneity. Did the author observe any effects on the measurement results?
Response 1: Thank you for your comments. We understand your concern so shows the real catheter for operation in the hospital in Figure R1(a). We cannot fix the length of the catheter so FBG should be fitted for 1.5 m in length. Figure R1(b) shows the 1.5 m of FBG but the Bragg grating part is very short as 1 cm which is located at the tip of FBG in Figure R1(c). Therefore, we attached the tip of FBG to our structure and the simulation was also concentrated in a few millimeters section. The distortion of FBG is only affected on the Bragg grating part (1 cm) so other parts are not affected by the wavelength signal.
Figure R1. (a) The commercial catheter with a length of 1.5 m, (b) the full length of 1.5 m fiber with FBG sensing part, (c) the FBG sensing part.
Point 2: The results in the new Fig.2(a-d) are very different from the previous one under the same simulation setting (0.5 N). For example, in the new Fig. 2a, the maximum stress is around 300 MPa but in the previous version, the maximum stress was around 20000 MPa. The author should double-check the results in Fig. 2 and make sure they are correct.
Response 2: Sorry for our mistake of scale. The previous scale is showing a high scale, but the actual force was smaller than the scale. Therefore, the new figure 2 is the correct numerical value.
Figure 2. Stress distributions for 0.5 N loading by ANSYS. (a) lateral force analysis for helical structure and (b) perforated pattern structure, (c) axial force analysis for helical structure and (d) perforated pattern structure; (e) three-degree conditions for force (0, 45, 90°) with the sensitivity of helical and perforated pattern structures. (n=3).
Point 3: They authors did not answer why they need to use three FBGs since the sensor is targeting vertical direction.
Response 3: Thank you for your comments. We showed each FBG has high sensitivity in a vertical direction with high repeatability in Figure 3. We suggested three FBGs can pursue higher accuracy and stability by average value through the LabVIEW program and even if one or two FBGs are damaged, the remaining FBG can still work.

Reviewer 4 Report
The authors improved the English writing.
The comments of the first review were answered in the response but still the message of the paper is unclear. The answers given do not help to demonstrate the value of this research in terms of a real clinical need. The papers that are cited to show the need e.g. [5] are in the field of robotic catheterization coming with completely different requirements.
The question for the costs of catheters is demonstrating, that the authors are not familiar with the application at all. (Central venous catheter is compared to cardiac ablation???)
Finally, the author's response has not even found its way into the paper itself, leading to a very limited improvement of this contribution. Please rewrite this paper with a high focus on correct scientific writing.
Author Response
Response to Reviewer 4 Comments
Remarks to the Author: The authors improved the English writing.
Point 1: The comments of the first review were answered in the response but still the message of the paper is unclear. The answers given do not help to demonstrate the value of this research in terms of a real clinical need. The papers that are cited to show the need e.g. [5] are in the field of robotic catheterization coming with completely different requirements.
Response 1: Thank you for your comments. As considered your comments, we agree to delete reference [5] in the revised manuscript. The robotic catheterization is different from our goal. Therefore, we added other references in the introduction parts.
Reference
- L. Di Biase; A. Natale; C. Barrett; C. Tan; C. S. Elayi; C. K. Ching; P. Wang; A. AL‐AHMAD; M. Arruda; J. D. Burkhardt, Relationship between catheter forces, lesion characteristics,“popping,”and char formation: experience with robotic navigation system. J. Cardiovasc. Electrophysiol. 2009, 20, 436-440.
Point 2: The question for the costs of catheters is demonstrating, that the authors are not familiar with the application at all. (Central venous catheter is compared to cardiac ablation???)
Response 2: Thank you for your comments. We probably used the cost expression confusingly. The detail of cost is only to the cost of adding a force sensor to the catheter, not the total cost required for the catheter procedure. Figure R1(a) shows a real catheter for using operation in the hospital and its cost is expensive. However, we applied to modify the tip parts with SS structure and FBG for the force sensor. We mentioned it for cost $200 in the manuscript.
Figure R1. (a) The commercial catheter with a length of 1.5 m, (b) the full length of 1.5 m fiber with FBG sensing part, (c) the FBG sensing part.
Point 3: Finally, the author's response has not even found its way into the paper itself, leading to a very limited improvement of this contribution. Please rewrite this paper with a high focus on correct scientific writing.
Response 3: Thank you for your comments. We modified our manuscript according to the reviewer’s comments.
